# T-shaped alignments integrating HIV-1 near full-length genome and partial pol sequences can improve phylogenetic inference of transmission clusters

August Guang[1]*, Casey W Dunn[2], Vlad Novitsky[3], Mark Howison[4], Rami Kantor[3]

**1** Center for Computation and Visualization, Brown University, Providence, Rhode Island, United States of America, **2** Ecology and Evolutionary Biology, Yale University, New Haven, Connecticut, United States of America, **3** Warren Alpert Medical School, Brown University, Providence, Rhode Island, United States of America, **4** Research Improving Peoples' Lives, Providence, Rhode Island, United States of America

* august_guang@brown.edu

## Abstract

Molecular epidemiology and HIV-1 transmission networks reconstruction can provide insights into transmission dynamics and inform public health strategies. Long HIV sequences, such as near full-length (nFL) genomes, can improve the accuracy of phylogenetic inference. However, relatively short pol sequences are still broadly used for inferring molecular HIV clusters. Whether a mix of long and short HIV-1 sequences can improve phylogenetic inference of molecular HIV clusters remains unknown.

We propose a flexible approach called T-shaped alignments that incorporates both nFL HIV-1 genomes and partial pol sequences, and investigate whether this approach improves phylogenetic reconstruction of molecular clusters. Under the assumption that clustering from 100% of long sequences is the most accurate, we obtained 1196 subtype B nFL HIV-1 sequences from the Los Alamos National Laboratory Database and a single-study subset, varied the proportion of long and short sequences in our T-shape alignments, systematically masked all non-pol regions with missing characters in proportional increments, and compared tree similarity and cluster inference among datasets.

With the full dataset, we found that when more than 50% of available sequences are nFL, the T-shaped alignment gradually yields results closer to the 100% n, with more and larger clusters identified. However, below the 50% threshold accuracy did not increase. Stringent bootstrap thresholds decreased cluster accuracy gaps but also decreased number of clusters found and mean cluster size. For the subset dataset, we found that the introduction of nFL sequences to the T-shaped alignment improves accuracy in clustering either after a 30% threshold or immediately depending on bootstrap choice.

**Data availability statement:** All data required to replicate the study's findings can be found here https://www.hiv.lanl.gov/content/sequence/HIV/mainpage.html. All analysis source code is available at https://github.com/dunnlab/hiv_wide.

**Funding:** This study was supported by grants from the National Institute of Allergy & Infectious Diseases of the National Institutes of Health (RO1AI136058, K24AI134359, P30AI042853) to RK and by an Institutional Development Award (IDeA) from the National Institute of General Medical Sciences of the National Institutes of Health (P20GM109035). The funders had no role in study design, data collection and analysis, decision to publish, or preparation of the manuscript.

**Competing interests:** The authors have declared that no competing interests exist.

Our new approach and results suggest that using T-shape alignments to mix HIV-1 sequences of different lengths can improve phylogenetic and clustering accuracy, with needed nFL proportion depending on analysis goals. The T-shape alignment provides a straightforward method for utilizing all available sequences to improve phylogenetic analysis.

## Author summary

We introduce and explore a novel approach to analyzing HIV-1 clustering through advanced genomic sequence analysis techniques. Unlike traditional molecular cluster inference methods that focus on widely available short segments of the virus's genetic code, our research leverages longer sequences, potentially offering a more detailed view of the virus's transmission patterns, paving the way for improved public health strategies and more effective efforts to control the spread of HIV-1.

Recognizing that obtaining long sequences from each individual diagnosed with HIV is not always feasible, our approach combines both short and long sequences, enabling us to utilize all available data effectively. This innovative method, which we term "T-shaped alignment,", allows integration of sequences of varying lengths without requiring new software, maximizing flexibility and efficiency in analysis.

Using publicly available global sequence data, we found that this method may improve accuracy in inferring HIV-1 molecular clusters when more than a certain proportion of the data consists of long sequences. The threshold differs between different datasets. This finding, which should be validated in regional HIV sequence datasets, may ensure better use of genetic data to understand and control HIV-1 transmission. Our work underscores the importance of sequence length proportions in genetic analysis, providing preliminary insights that may be relevant for future public health initiatives.

## Introduction

The use of molecular epidemiology in inferring transmission clusters and shared characteristics between individuals with HIV-1 has been critical to providing insights into disease transmission dynamics and informing public health strategies [1–3]. Molecular epidemiology defines clusters of infections through molecular sequencing of viruses and other pathogens, utilizing metrics such as genetic distance and phylogenetic bootstrap. While public health contact tracing methods aim to provide known personal connection histories, molecular epidemiology can infer partial or complete transmission chains through viral sequence similarities and reconstructing viral phylogenies [4]. Researchers have traditionally relied on HIV-1 partial pol sequences to construct phylogenetic trees and infer transmission clusters as they contain

substantial phylogenetic signal [5], can be accurately used for HIV-1 transmission reconstruction [1], and are available through routine drug resistance testing in clinical care [6].

The development of long-read and next generation sequencing (NGS) covering near-full length (nFL) HIV-1 genomes have made them increasingly available for use in phylogenetic analysis and clustering [7,8]. Studies using nFL HIV-1 genomes have demonstrated improved accuracy when using long nFL genomes over short partial pol sequences in both phylogenetic and clustering analyses [9–11]. While using nFL genomes may be desirable, it may not be feasible in most settings. Many individuals previously diagnosed with HIV may be virally suppressed, have transferred care, or may not be interested in providing samples for sequencing, and resources might be limited. Importantly, combining partial pol sequences with newly generated nFL sequences in a single alignment for phylogenetic inference can present technical challenges, and trimming sequencing to the shortest length prior to phylogenetic inference is common.

A flexible approach that can incorporate both nFL HIV-1 genome sequences and partial pol sequences to possibly improve transmission cluster inference would be greatly advantageous because of the additional resolution that nFL genome sequences may provide. One such approach is to treat all partial pol sequences in the alignment as having missing characters for the rest of the genome, while still using the non-pol data, rather than only using pol sequences from the whole dataset. We refer to this as a T-shaped alignment, where, if sequences were sorted by completeness, the top of the T is the nFL genome sequences and the stem of the T is the pol-only sequences. Alignment methods that deal with sequence length heterogeneity in this way exist already, typically by computing a backbone alignment with the nFL sequences and then aligning fragments to the backbone alignment. [12–14] Standard phylogenetic methods can incorporate the T-shape alignment, bypassing the need for new software implementations. An additional advantage of this approach is that any length sequence from any overlapping regions of the genome can be incorporated into this alignment. To the best of our knowledge, no prior studies addressed such heterogeneous HIV-1 alignments.

We investigated whether the T-shape alignment approach would improve HIV phylogenetic reconstruction and molecular cluster analysis by looking into alignments with different proportions of nFL HIV-1 genomes and partial pol sequences from publicly available data, under the assumption that a 100% nFL genome alignment would provide the most accurate results. While the impact of different kinds of missing data on phylogenetic inference is a continuous area of research [15, 16], generally increasing data in both characters and taxa has been found to improve phylogenetic tree resolution [17,18], which suggests that the T-shaped alignment should improve HIV phylogenetic inference and clustering, potentially better informing public health.

However, we found that simply adding nFL genome sequences into the alignment does not always lead to increased accuracy in both the phylogenetic tree topology and cluster inference, and in fact appears to be more misleading at lower mixture proportions. Instead, the topology and cluster inference gradually improves after a majority of sequences in the alignment are nFL genomes.

## Materials and methods

We downloaded a filtered nFL genome alignment of all HIV-1 subtype B sequences published through the end of 2018 from the Los Alamos National Laboratory HIV database (full dataset) as our initial input [19] (Fig 1A) and assessed hypermutation in the alignment with HYPERMUT [20]. We found that out of the 1196 sequences, 4 were hypermutated. As this was a minimal number, we kept all sequences. To address potential biases arising from dataset heterogeneity and sequence source, we additionally created a subset of the filtered nFL genome alignment comprising of the 116 subtype B nFL genomes sequenced in [21] (subset dataset), selected because it is the largest subset of sequences from within a single study and contains geographic linkage information as well. This subset dataset served as a sensitivity analysis.

For both the full alignment dataset and the subset dataset, we created 100 alignment samples by shuffling the order of the taxa in the alignment randomly with a different seed each time (Fig 1B). For each sample, we created 13 datasets with

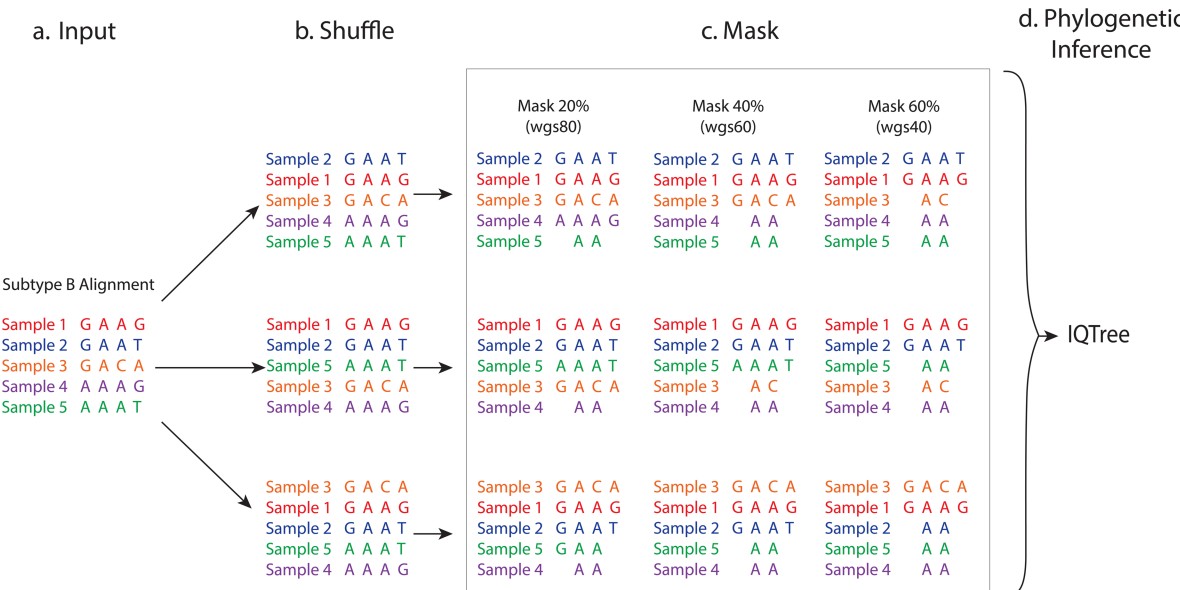

**Fig 1. Workflow for alignment dataset generation and phylogenetic analysis.** (A) The process begins with a downloaded, filtered near full-length (nFL) HIV-1 subtype B genome alignment as input. (B) To generate multiple alignment samples, the order of taxa within the input alignment is randomly shuffled. (C) For each shuffled sample, various datasets are created by masking specific genomic regions outside the protease-reverse transcriptase (PRRT) region, using HIV-1 HXB2 coordinates to introduce gaps. (D) The processed alignment datasets are then used as input for IQTree to generate maximum likelihood phylogenies.

different mixtures of nFL (referred to as wgs for whole genome sequences) and pol sequences by using the HIV-1 HXB2 (Genbank accession number K03455) genome coordinates to replace (mask) all characters outside of the protease-reverse transcriptase (PRRT) region with gaps according to the mixture of pol specified (Fig 1C). Eleven of the datasets had mixtures in increments of 10% of the taxa being nFL in the alignment starting from 0%, i.e. from 10% up to 100%, while 2 of the datasets were were mixtures of 95% nFL and 5% pol and 99% nFL and 1% pol, respectively. Alignments were labeled by the proportion of nFL sequences in the mixture, i.e. wgs40 represents 40% wgs and 60% pol sequences. This created 13,000 alignment datasets total.

Each alignment as well as the 100% whole genome alignment (wgs100) were run through the maximum likelihood phylogenetic inference program IQtree [22] to generate phylogenies with bootstrapping (Fig 1D). The IQTree algorithm treats gaps and missing characters as having no information, similar to most other maximum likelihood-based software such as RAxML and PhyML, and thus a site with missing taxa will have the same likelihood as if the missing taxa were not there. The midpoint clustering algorithm with bootstrapping thresholds 70, 80, 85, 90, 95, 99 was used to identify clusters on all phylogenies [23]. Genetic distance was not used due to the differing lengths of the sequences in the analysis and lack of established threshold for molecular clusters based on the entire HIV genome sequences.

For each sample we compared phylogenies and clustering between the different mixture datasets and wgs100 under the assumption that the wgs100 phylogeny and clustering was the most accurate and thus could act as the reference results. For phylogenies we evaluated tree similarity to the wgs100 reference by computing the Robinson-Foulds distances [24] between wgs100 and all other mixture trees. We also ran a one-tailed paired t-test on the Robinson-Foulds distances.

The Robinson-Foulds distances give us an idea of how similar mixture trees and the wgs100 tree are, but not how similar mixture trees are to other mixture trees. To further visualize and assess the similarity between phylogenies inferred from the different mixtures, we computed the Clustering Information Distance from the R package TreeDist as the tree

distance metric between all pairs of trees, based on the recommendations outlined in [25], then projected the distance matrix onto 2D with a Principal Coordinates Analysis (PCoA).

For clustering we computed the mean number of clusters and mean cluster size for each mixture dataset at each bootstrap threshold to understand how cluster attributes changed as mixtures and bootstrap thresholds changed. We evaluated clustering congruence through two metrics: the Adjusted Mutual Index (AMI) [26] and the overlap coefficient. AMI was chosen due to the unbalanced cluster sizes [27], while overlap cofficient was chosen due to its ability to capture subset relationships as it directly measures the proportion of shared elements relative to the smaller of the two clusters. Both metrics were computed between 100% wgs and all other mixtures at the same bootstrap, i.e. clustering between 100% wgs and mixtures at 80 bootstrap were compared to each other but not to clustering at 90 bootstrap. For AMI, clusters were defined as matching if they shared all of the same members, and non-matching otherwise, including cases where one cluster was a superset of another. An AMI closer to 1 represented higher congruence, while an AMI closer to 0 represented lower congruence. For the overlap coefficient, a coefficient closer to 1 meant that more members of one cluster are also all members of the other cluster, while a coefficient closer to 0 meant that almost no members of either cluster are members of the other.

To investigate our first hypothesis for trends in phylogeny and clustering as mixture proportions changed, we looked at solely the wgs50 phylogeny and computed the number of clusters that had all pol sequences, all wgs sequences, and a mix of both. To investigate our second hypothesis, we masked all gp120 regions from the alignment and then reran the same methodology as above, i.e. IQTree followed by bootstrap-based clustering at different thresholds along with computing Robinson-Foulds distances and AMI. To investigate our third hypothesis, we extracted substitution model parameters from each prior IQTree sample run and created a boxplot of the parameters. We then reran the same methodology as above on all samples, but with the substitution model parameters from the wgs100 tree instead.

All analysis source code is available from: https://github.com/dunnlab/hiv_wide.

## Results

### Tree similarity increases after at least 40-50% of sequences are near full-length genomes

We assessed similarity between all pol- and whole genome sequence (wgs) mixture phylogenies and the wgs100 trees by plotting the normalized Robinson-Foulds distances, a measure of topological similarity between phylogenetic trees (Figs 2 and 3). For the full dataset, the wgs50 through wgs99 mixtures show decreasing distance to wgs100, indicating that their topologies are closer to wgs100 as the proportion of wgs grow (Fig 2). However, pol through wgs40 have varying distances to wgs100 and do not necessarily increase. The one-tailed paired t-test showed that at wgs50 the distance to wgs is closer than pol to wgs (p-value 4.791e-14), but mixtures with lower wgs proportions are not. Additionally, the spread of distance to wgs100 appeared to increase as the proportion of wgs increased. For the subset dataset, the wgs40 through wgs99 mixtures show decreasing distance to wgs100 (Fig 3), and the one-tailed paired t-test showed that at wgs40 the distance to wgs is closer than pol to wgs (p-value 0.002534), but mixtures with lower wgs proportions are not. As the proportion of wgs grows, the variation in distances to wgs100 also increases (larger whiskers and boxes), likely due to the longer sequences with more variable regions (e.g. *env*) filling in gaps and thus leading to more variation at those sites.

Plotting mixtures onto the two-dimensional Principal Coordinates Analysis (2D PcoA) projection, percent variance explained over both axes (the measure of how much the first 2 principal coordinates contribute to variation in the distance matrix) was fairly low (Fig 4, 10.98% x-axis+3.25% y-axis; Fig 5, 24.4% x-axis+3.63% y-axis). For both the full dataset and the subset dataset, along the first (X) variance dimension, similarity to wgs100 directly increased as proportion of wgs sequences increased, (moving from right to left on the X-axis) suggesting that there is a direct relationship between increasing the proportion of wgs sequences and similarity to the wgs100 tree. Along the second variance dimension (Y-axis), similarity to wgs100 did not directly increase as proportion of wgs sequences increased, with the points forming

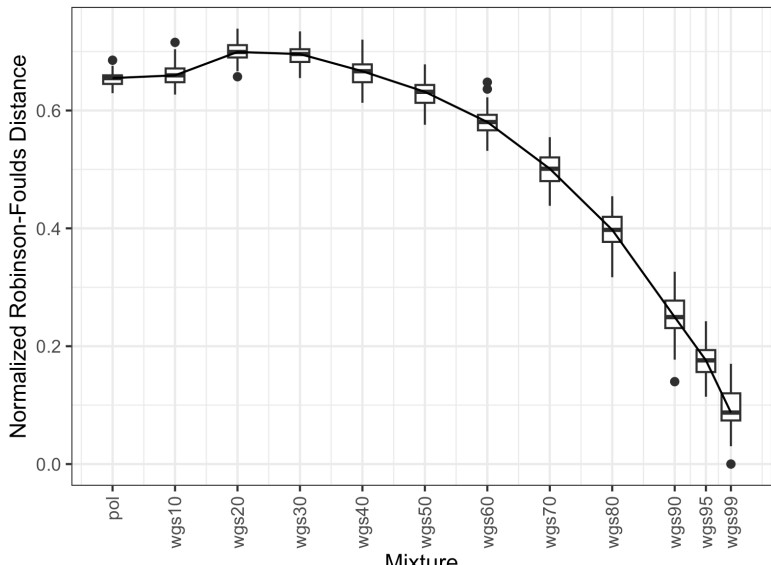

**Fig 2**. **Phylogenetic distance between partial pol-wgs mixtures and nFL genome for full dataset.** Boxplot of Robinson-Foulds distance (Y axis) between maximum-likelihood phylogenies generated from different mixtures of pol and wgs sequences (X axis; e.g. wgs60 means 60-40 wgs-pol, respectively) to the phylogeny generated from a nFL genome alignment. Boxplots contain the first and third quartiles as well as the mean, with the vertical lines indicate the range. Points represent individual outlier distances.

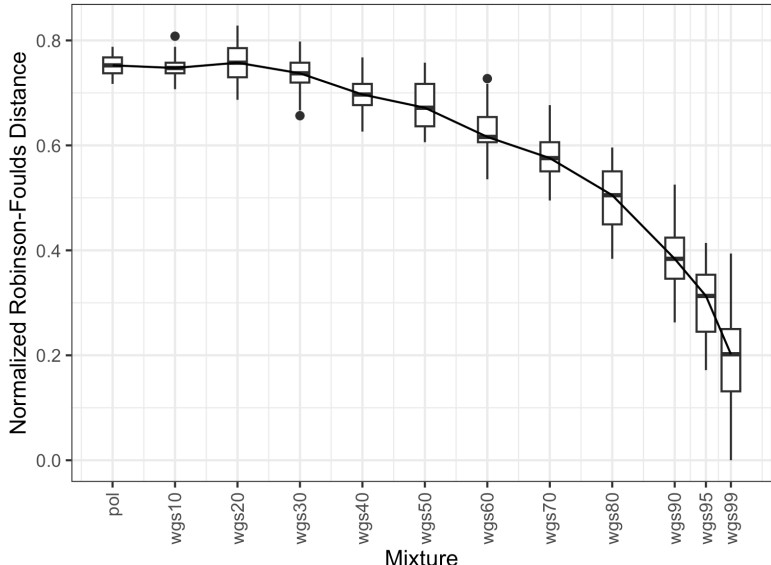

**Fig 3**. **Phylogenetic distance between partial pol-wgs mixtures and nFL genome for subset dataset.** Boxplot of Robinson-Foulds distance (Y axis) between maximum-likelihood phylogenies generated from different mixtures of pol and wgs sequences (X axis; e.g. wgs60 means 60-40 wgs-pol, respectively) to the phylogeny generated from a nFL genome alignment. Boxplots contain the first and third quartiles as well as the mean, with the vertical lines indicate the range. Points represent individual outlier distances.

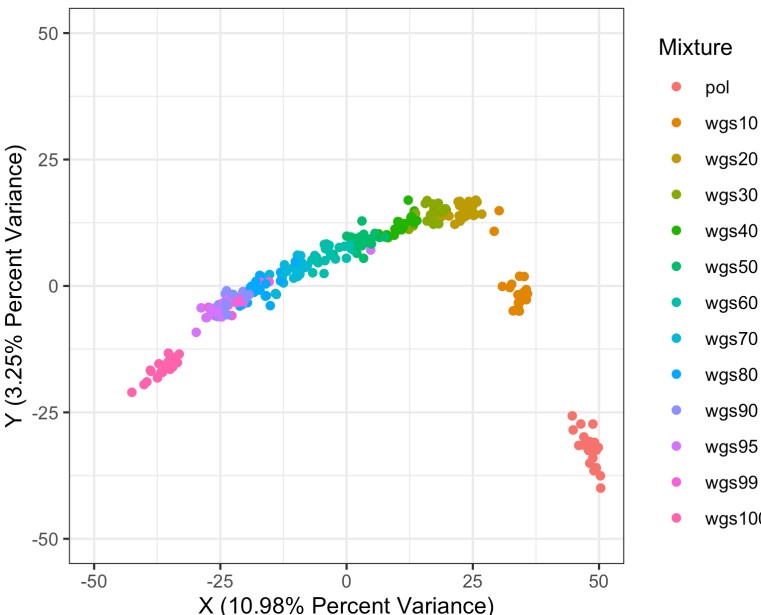

**Fig 4**. **Principle Coordinates Analysis (PCoA) projected onto 2 dimensions between pol-wgs mixtures and nFL genome, full dataset** X-axis is the first principle coordinate which explains the largest amount of data change, Y-axis is the second principle coordinate which explains the second largest amount of data change. Distances on axes summarize variability and are relative to the samples plotted and colors indicate different sequence mixtures according to the legend. Along the first (highest) variance dimension, different mixtures varied directly from pol to wgs, while along the second variance dimension, pol and wgs were closer to each other than to wgs30.

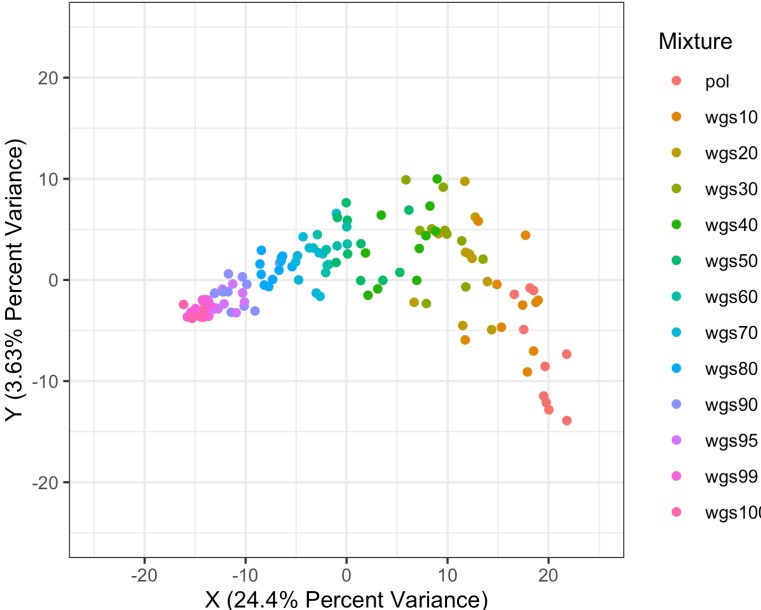

**Fig 5**. **Principle Coordinates Analysis (PCoA) projected onto 2 dimensions between pol-wgs mixtures and nFL genome, subset dataset** X-axis is the first principle coordinate which explains the largest amount of data change, Y-axis is the second principle coordinate which explains the second largest amount of data change. Distances on axes summarize variability and are relative to the samples plotted and colors indicate different sequence mixtures according to the legend. Along the first (highest) variance dimension, different mixtures varied directly from pol to wgs, while along the second variance dimension, most between-sample distances fell between 10 and -10. pol through wgs30 had higher spreads with less distinct clustering compared to the rest of the mixtures.

a u-shape instead. Additionally, for the full dataset trees from different mixtures clearly clustered and separated on the 2D projection. For the subset dataset, trees from different mixtures did not clearly cluster at lower proportions of wgs. This suggests that while increasing the proportion of wgs sequences is the dominant factor and does linearly increase similarity to wgs100, there is a second non-linear factor impacting phylogenetic distance.

## Cluster numbers and mean cluster size decrease until wgs50, then gradually increase

For each bootstrap threshold and dataset mixture, we computed the mean number of clusters and mean cluster size averaged over the dataset. For the full dataset, the number of clusters and mean cluster size decreases as bootstrap thresholds increase (Fig 6 and S1 Table). At bootstrap thresholds of 90, 95 and 99 the mean number of clusters decreases by a large degree as wgs proportion increases up to wgs50, then increases as wgs proportion increases from there. At bootstrap thresholds of 70, 80 and 85 the mean number of clusters decrease slightly or stay about the same as wgs proportion increases up to wgs50, then increase slightly as wgs proportion increases from there. Additionally, mean cluster size slightly decreases as wgs proportion increases up to wgs50, then increases from there.

For the subset dataset, at bootstrap thresholds of 90, 95, and 99 the mean number of clusters stays around the same until wgs50, then increases as wgs proportion increases from there (Fig 7 and S2 Table). At bootstrap thresholds of 70, 80, and 85 the mean number of clusters appears to decrease slightly or stay about the same as wgs proportion increases up to wgs50, then increase slightly. Mean cluster size appears to increase steadily as wgs proportion increases. This provides some indication that increasing wgs sequence proportion after wgs50 can lead to detecting more and larger clusters.

## Clustering improves after wgs50 for full dataset, and depends on bootstrap threshold for subset dataset

For each bootstrap threshold, we plotted the Adjusted mutual information (AMI; a measure to evaluate clustering similarity that accounts for chance) between clustering from each mixture phylogeny and the wgs phylogeny (Figs 8 and 9). For the full dataset, at all bootstrap thresholds, there is no linear relationship between including wgs and closeness of clustering results to the wgs tree (Fig 8). Wgs20 has the lowest AMI across all bootstrap thresholds, indicating that the transmission clusters inferred with the wgs20 tree are least congruent with wgs100. Pol, wgs10, wgs30, wgs40, and wgs50 all share similar AMIs. After wgs50 the congruence to wgs100 increases as proportion of wgs sequences increases, with higher similarity to wgs100 than to pol based on the t-test. This indicates that including wgs sequences can increase clustering accuracy, but only after at least 50% of sequences are wgs.

For the subset dataset, at bootstrap thresholds of 70 and 99, AMI increases with the addition of wgs sequences, while at the other bootstrap thresholds, AMI increases after wgs30 based on t-test (Fig 9). At all other bootstrap thresholds, the AMI increases immediately. We note that the AMI is relatively low across all bootstrap thresholds – this result is expected because of the exponentially large number of possible clusters.

We additionally measured the overlap coefficient due to its ability to capture subset relationships where one cluster is contained within another. We plotted the proportion of wgs100 clusters recovered at two different overlap coefficient thresholds: 50% and 66% (Figs 10 and 11). If a cluster in a mixture phylogeny had an overlap coefficient of at least 50% or 66% with a cluster in the wgs100 tree, then we considered the cluster recovered in the mixture phylogeny. We found that for the full dataset (Fig 10), the pattern of improvement followed as previously, with at least 50% of sequences needing to be nFL before the proportion of recovered wgs100 clusters improved in accuracy relative to an all pol phylogeny. For the subset dataset (Fig 11), the increase in proportion of recovered wgs100 clusters varied depending on bootstrap or overlap coefficient threshold, but generally increased steadily after 30%, if not before. With this metric the proportion of clusters recovered was generally higher, with recovery proportion approaching 100% at all bootstrap and overlap coefficient thresholds for the subset dataset.

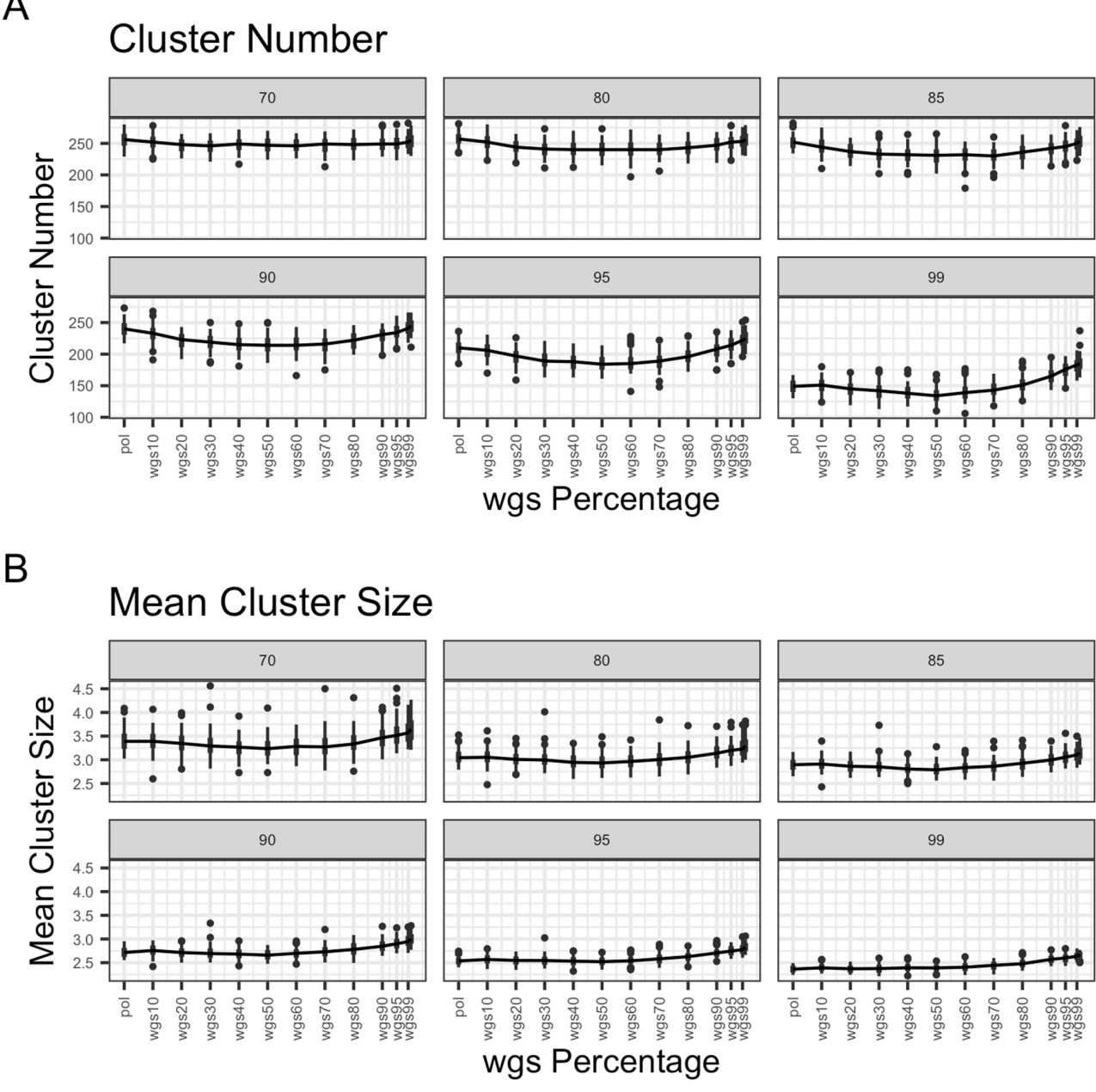

**Fig 6. Scatter and line plots of cluster number and mean cluster size as wgs percentage in mixture increases, full dataset.** The figure demonstrates (A) the number of clusters (Y axis) for each pol-wgs mixture dataset (X axis) at different bootstrap thresholds (colors; legend) and (B) mean cluster size (Y axis) for each pol-wgs mixture dataset (X axis) at different bootstrap thresholds (colors; legend). Trend lines with error bars are drawn for each bootstrap threshold as well as individual sample points. Sample points were plotted with a jitter to better display the spread across cluster number and mean cluster sizes, but only represent mixtures in proportions of 10, i.e. wgs0 (pol), wgs10, wgs20, wgs30, etc.

## Possible explanations

We looked into three possible explanations for why combining long nFL HIV-1 sequences and short pol sequences does not always improve clustering and in lower proportions of wgs can even lead to less accurate results for large datasets with many unrelated sequences such as in the full dataset.

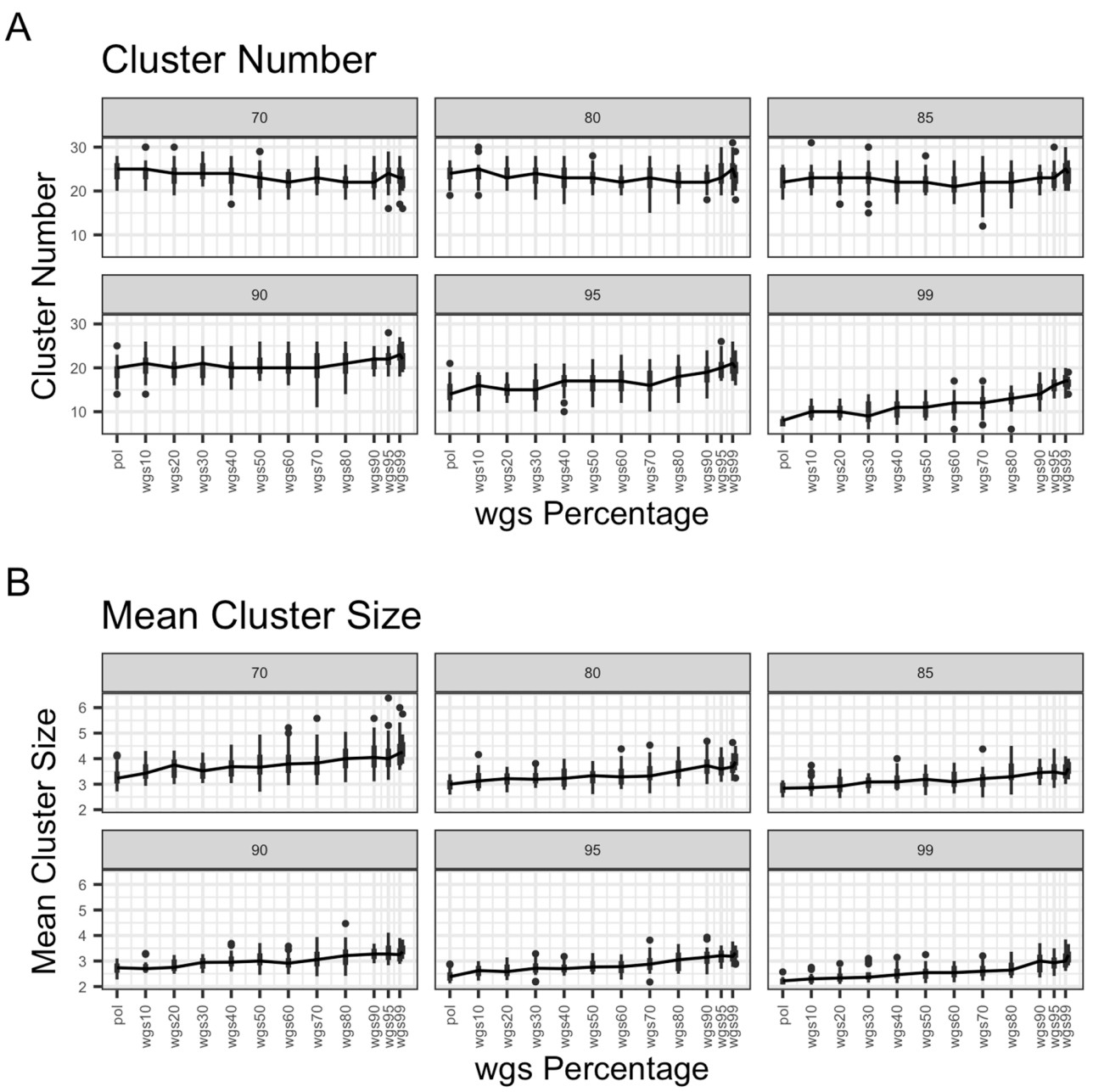

**Fig 7**. **Scatter and line plots of cluster number and mean cluster size as wgs percentage in mixture increases, subset dataset.** The figure demonstrates (A) the number of clusters (Y axis) for each pol-wgs mixture dataset (X axis) at different bootstrap thresholds (colors; legend) and (B) mean cluster size (Y axis) for each pol-wgs mixture dataset (X axis) at different bootstrap thresholds (colors; legend). Trend lines with error bars are drawn for each bootstrap threshold as well as individual sample points. Sample points were plotted with a jitter to better display the spread across cluster number and mean cluster sizes, but only represent mixtures in proportions of 10, i.e. wgs0 (pol), wgs10, wgs20, wgs30, etc.

**Pol and WGS do not cluster together.** One hypothesis about the reasoning for the incongruence in mixtures is that pol-only tips could spuriously group together, and wgs tips could spuriously group together. We found a slight bias towards pol sequences grouping together. We calculated the mean proportion of pol sequences in clusters at each mixture proportion, with the expectation that the proportion of pol sequences in clusters should be equal to the proportion

**Fig 8. Adjusted mutual information (AMI) boxplot between phylogenetic clustering of different mixtures of pol and wgs sequences and all wgs, full dataset.** The figure demonstrates six panels for different bootstrap thresholds (indicated in gray in the top bars). Boxes in panels indicate the central 50% (top and bottom of boxes) and the median (thicker black line in boxes) of the AMI (Y axes) for a given set of pol-wgs mixture datasets (X axes). Whiskers indicate the range, and dots indicate outliers. Higher values of AMI indicate higher congruency in cluster inference between the mixture dataset and the wgs100 tree at the given bootstrap threshold.

of pol sequences in the phylogeny, but found that for every mixture proportion it was higher. For example, for wgs50, the mean proportion of pol sequences in clusters was around 0.64, suggesting a bias towards pol sequences in clusters.

**Fastest evolving sites do not contribute to phylogeny.** Another hypothesis was that the fastest evolving and polymorphic sites outside of pol skewed the phylogeny due to their extremely fast mutation rates leading more wgs tips to pull together due to long branch attraction [28,29]. In order to investigate this, we masked the gp120 region from all phylogenies and reproduced the analysis described above. However, we did not see any difference compared to using the whole genome.

**Substitution model parameters do not affect phylogeny.** The third hypothesis was that due to the missing data, the model would be unable to infer the correct substitution model parameters for the phylogenetic tree inference, which would then impact both the phylogeny and the clustering results. To investigate this, we first looked at the substitution model parameters inferred from IQTree; the parameters differed significantly between the wgs alignment and the mixture datasets. The parameter estimates from the pol alignment in these cases were very different from the mixtures, with the pattern of estimates becoming closer to wgs as the proportion of wgs increased (Fig 12, S1 Figure, and S2 Figure). This suggests that when there is lots of missing data, the model is unable to estimate parameters for those regions appropriately. The pol alignment having very different parameters is unsurprising as pol evolves quite differently from the other regions. However, when we reran IQTree fixing the parameter estimates to those derived from wgs to see if the substitution model parameters affected the phylogeny, we found that they did not.

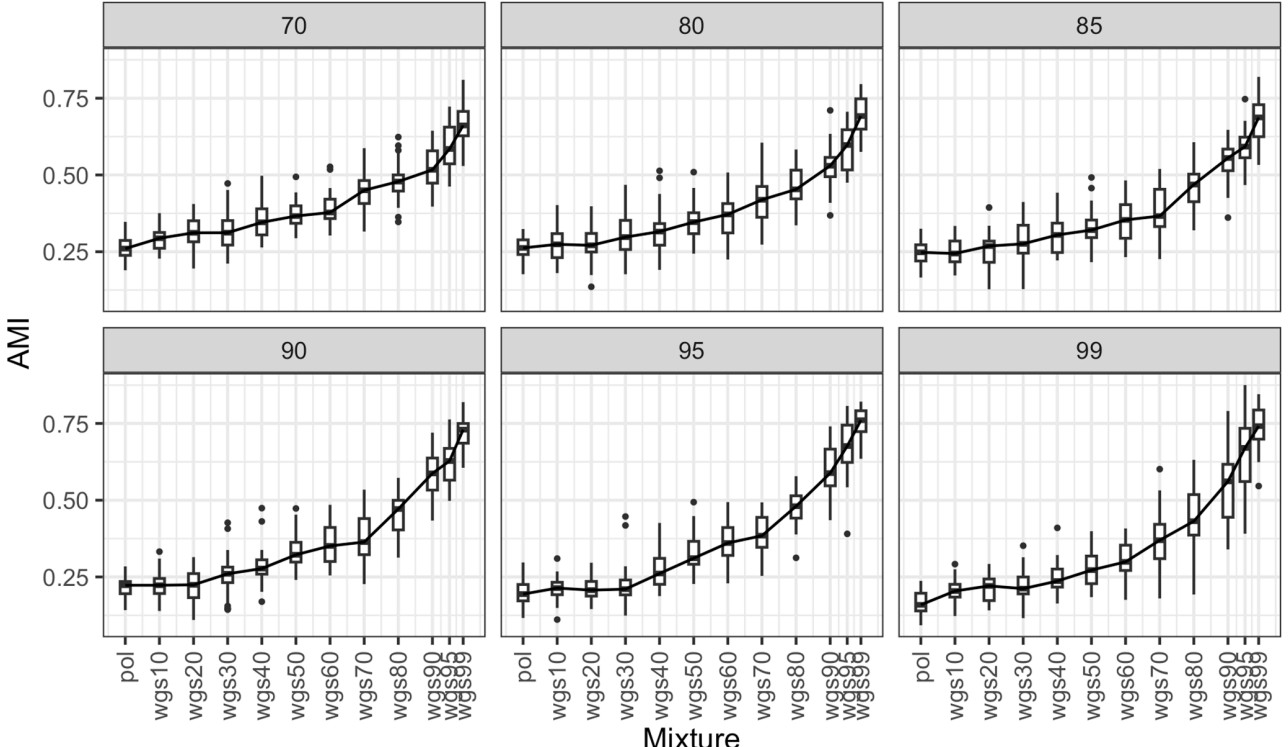

**Fig 9. Adjusted mutual information (AMI) boxplot between phylogenetic clustering of different mixtures of pol and wgs sequences and all wgs, subset dataset.** The figure demonstrates six panels for different bootstrap thresholds (indicated in gray in the top bars). Boxes in panels indicate the central 50% (top and bottom of boxes) and the median (thicker black line in boxes) of the AMI (Y axes) for a given set of pol-wgs mixture datasets (X axes). Whiskers indicate the range, and dots indicate outliers. Higher values of AMI indicate higher congruency in cluster inference between the mixture dataset and the wgs100 tree at the given bootstrap threshold.

## Discussion

We assessed a T-shape alignment approach that can combine sequences of different lengths into phylogenetic analyses and cluster inference and does not require new software implementations. Applying this approach to HIV-1 subtype B sequences, we assessed its potential advantages over conventional short-only or long-only sequence datasets. We found that for large datasets with many unrelated sequences, at lower proportions of nFL genome sequences, the T-shape alignment does not seem to offer any improvement over the use of just pol. For smaller datasets, the T-shape alignment can offer improvement over the use of just pol, but the proportion of nFL genome sequences needed depends on choice of bootstrap threshold as well. The difference between the large dataset and smaller dataset could be due to the larger dataset coming from different studies and techniques, while the smaller study comes from a singular region.

When over 50% of sequences are whole genome, combining whole genome and pol sequences into a T-shape alignment generates phylogenetic trees and clusters that are gradually closer to those from a 100% whole genome alignment with increasing proportion of nFL genome sequences in the alignment. Assuming that wgs has benefits over sequencing a single gene, this suggests that mixing sequences of different lengths can improve phylogenetic and clustering accuracy contingent on reaching a nFL proportion threshold. The nFL proportion threshold depends on both dataset features and on bootstrap threshold. These results are somewhat unexpected, as the phylogenetic literature suggests adding characters may improve tree resolution [17]. Contrary to the conventional belief that increasing data, both in characters and taxa, invariably enhances phylogenetic tree resolution, our findings underscore the nuanced dynamics of incorporating diverse sequence types.

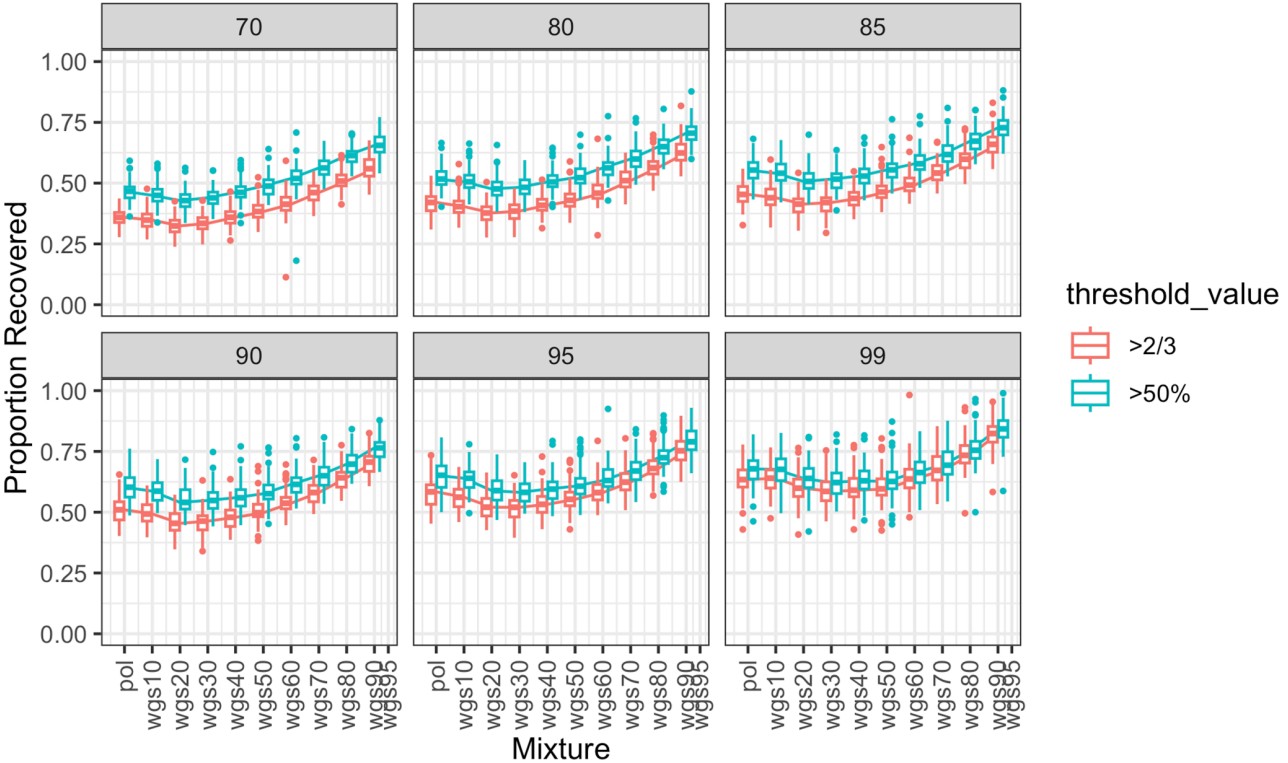

**Fig 10. Proportion of clusters recovered between different mixtures of pol and wgs sequences and all wgs based on overlap coefficient, full dataset.** The figure demonstrates six panels for different bootstrap thresholds (indicated in gray in the top bars). Boxes in panels indicate the central 50% (top and bottom of boxes) and the median (thicker black line in boxes) of the proportion of clusters recovered (Y axes) for a given set of pol-wgs mixture datasets (X axes). Whiskers indicate the range, and dots indicate outliers. Colors indicate overlap coefficient threshold to consider a cluster recovered. Higher values on the Y-axis indicate higher congruency in cluster inference between the mixture dataset and the wgs100 tree at the given bootstrap threshold.

We explored possible explanations for this result. We looked at excluding fast evolving genes, which has been shown to produce different phylogenies in other contexts [30], but we did not see any difference compared to using the whole genome. We also looked at whether changing substitution model parameters would affect the results, but did not see any effect, although there is some prior phylogenetic literature on the limited impact of different substitution model parameters in other systems [31]. Finally, we assessed whether pol sequences may be clustering with other pol sequences, which would subsequently bias the clustering results as well. We found a slight bias towards pol sequences grouping together, suggesting a possible reason behind needing a specific nFL proportion threshold and an avenue for future investigation.

Our findings have potential implications for HIV-1 phylogenetic and clustering studies. Specifically, in molecular HIV cluster analyses, mixing nFL HIV-1 genome sequences with partial pol sequences may be advantageous only if the proportion of nFL genome sequences exceeds a specific threshold. The threshold appears to differ based on dataset features. In analyses of large datasets with many unrelated sequences, improvement occurs when the majority of sequences are nFL, but in more local settings a lower proportion of nFL sequences may be sufficient. Future work will look into how the T-shaped alignment performs on a Rhode Island regional dataset, as regional clustering in a real setting may provide greater insight into transmission networks that could be relevant for informing public health interventions [32–35].

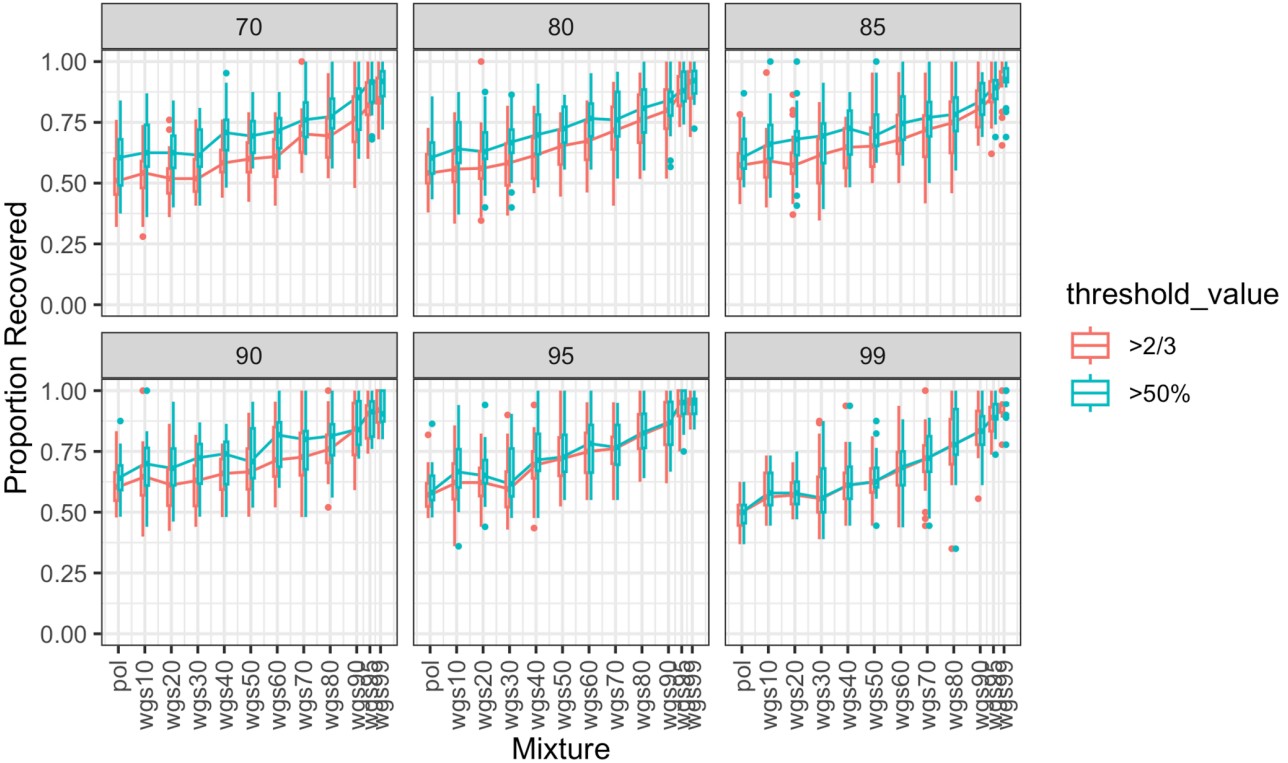

**Fig 11. Proportion of clusters recovered between different mixtures of pol and wgs sequences and all wgs based on overlap coefficient, sub-set dataset.** The figure demonstrates six panels for different bootstrap thresholds (indicated in gray in the top bars). Boxes in panels indicate the central 50% (top and bottom of boxes) and the median (thicker black line in boxes) of the proportion of clusters recovered (Y axes) for a given set of pol-wgs mixture datasets (X axes). Whiskers indicate the range, and dots indicate outliers. Colors indicate overlap coefficient threshold to consider a cluster recovered. Higher values on the Y-axis indicate higher congruency in cluster inference between the mixture dataset and the wgs100 tree at the given bootstrap threshold.

Bootstrap thresholds further influence clustering outcomes. Stricter thresholds reduce the gap in cluster accuracy between mixed datasets and wgs100, likely due to the conservative nature of bootstrap support [36], and also decrease the number of clusters and size of clusters found. For smaller datasets with many sequences from the same region, boot-strap threshold affects the proportion of nFL sequences needed to improve accuracy as well. At the most relaxed and strictest bootstrap thresholds (70 and 99), the addition of nFL sequences into a T-shaped phylogeny always improves accuracy, while at thresholds in-between, at least 30% of nFL sequences are needed to improve accuracy. The number of clusters and mean cluster size are also influenced by the proportion of wgs sequences and bootstrap thresholds, with increases in number of clusters and mean cluster size after wgs50. These trends suggest that increasing wgs proportions facilitates the detection of larger and potentially more meaningful clusters, but depends on reaching a certain threshold of nFL sequences. Bootstrap threshold choice is often dependent on the stated public health or scientific goals, with stricter thresholds suggested for rapidly growing clusters or low viral diversity epidemics, and more relaxed thresholds suggested for routine public health tracking and longer time periods [37]. Our findings from varying the bootstrap threshold reinforce these suggestions. We further suggest that regardless of bootstrap threshold choice, incorporating at least 50% of wgs sequences may provide an improvement in cluster accuracy, number, and size of clusters detected. The impact of this approach on regional HIV epidemics and public health interventions is yet to be determined.

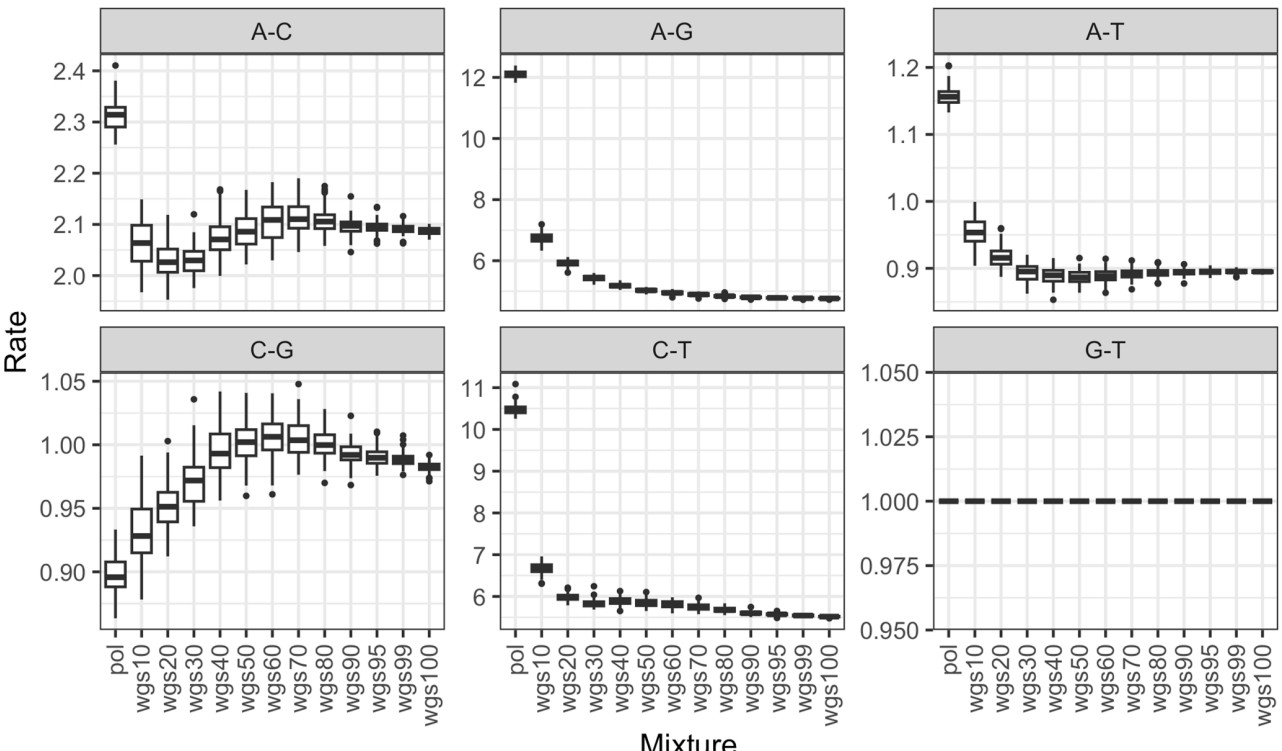

**Fig 12**. **Impact of different substitution rate category parameters on cluster inference in different mixture datasets.** The figure explores the impact of different substitution rates in our choice of substitution model for IQTree (GTR+F+I+G4), with the 6 different substitution rates (A-C, A-G, A-T, C-G, C-T, G-T, gray bar on top of each panel) on the Y axis, ordered by the different mixture datasets (X axis). Boxes in panels indicate the central 50% (top and bottom of boxes) and the median (thicker black line in boxes) of the substitution rate, whiskers indicate the range, and dots indicate outliers.

One possible explanation for our results we did not explore is that the clusters fluctuate in size and composition but maintain a degree of consistency depending on the bootstrap threshold chosen. A preliminary analysis of a set of 10 randomly chosen clusters revealed that about half of the clusters simply lost or gained members as bootstrap threshold varied, but that the other half completely disappeared or reappeared. Further analysis following the categorizations and cluster typing outlined in [33] could shed further light on this.

Our study comes with limitations. First, we did not choose a genetic distance threshold when performing our analyses due to the heterogeneity of sequence types as well as lack of studies validating genetic distance thresholds for nFL genome sequences. Genetic distance thresholds are commonly used in pol based phylogenetic analyses, particularly as related to defining public health associated clusters, but choosing the relevant threshold is an active area of research [37–39]. Recent methods such as AUTO-TUNE [40] that systematically select distance thresholds based only on the sequencing data could be applied to our data in order to generate a set of distance thresholds for different mixture proportions that create more comparable clustering with consistent and desirable properties such as ratio of largest cluster to second-largest cluster size. It is also possible that within this set of distance thresholds, one in nFL genome sequences exists that makes clustering results congruent regardless of pol and nFL genome mixture level or makes results more congruent to 100% nFL genome alignments. We did perform an initial investigation into this approach, but did not find any clear distance thresholds. Second, recent work also suggests that bootstrap and genetic distance thresholds are heuristics and proposes using sampling dates to better estimate emerging clusters [41]. As the dataset used in this study does not contain precise sequencing or sampling dates, repeating this analysis with a dataset with available sample

or sequencing dates would shed further light on these questions. Lastly, we used the wgs100 dataset, and the number and sizes of clusters identified in it, as the gold standard; however, the accuracy of this approach still needs to be determined.

In conclusion, we introduced T-shape alignment as an approach to leverage sequences of different lengths for phylogenetic cluster inference without requiring new software implementations. Practically, results suggest feasibility of integrating such datasets into analyses of HIV molecular clusters, but recommend a cautious approach - their inclusion may be justified only when the majority of sequences are nFL genomes, and should take bootstrap threshold selection into careful consideration. Since phylogenetic trees enhance pure genetic distance-based methods and supplement contact tracing in public health settings, being able to use nFL HIV-1 genome sequences may provide additional measures of cluster certainty when performing phylogenetic-based analyses.

## Supporting information

**S1 Table. Mean number of clusters and cluster size at different bootstrap thresholds and wgs proportions, full dataset.** Each row represents a different mixture of wgs and pol sequences, going from 100% pol to 100% wgs. Each column represents a bootstrap threshold for clustering, with the subcolumns indicating the mean number of clusters and mean cluster size found at that bootstrap. Cluster number was averaged over all samples for a given mixture, and cluster size was averaged over both all cluster sizes in a given sample, and over all samples.
(DOCX)

**S2 Table. Mean number of clusters and cluster size at different bootstrap thresholds and wgs proportions, subset dataset.** Each row represents a different mixture of wgs and pol sequences, going from 100% pol to 100% wgs. Each column represents a bootstrap threshold for clustering, with the subcolumns indicating the mean number of clusters and mean cluster size found at that bootstrap. Cluster number was averaged over all samples for a given mixture, and cluster size was averaged over both all cluster sizes in a given sample, and over all samples.
(DOCX)

**S1 Fig. Rates of different base frequencies from different mixture datasets.** The figure explores the impact of the 4 different base frequency proportions (pi(A), pi(C), pi(G), pi(T), gray bar on top of each panel) in our choice of substitution model for IQTree (GTR+F+I+G4). Proportion is on the Y-axis, and the different mixture datasets are ordered from *pol* to wgs100 on the X axis. Boxes in panels indicate the central 50% (top and bottom of boxes) and the median (thicker black line in boxes) of the proportions, whiskers indicate the range, and dots indicate outliers. There is a monotonic relationship between relative rate values and wgs proportion beginning with wgs10, when wgs sequences are introduced.
(TIFF)

**S2 Fig. Rates of different gamma relative rate categories from different mixture datasets.** The figure explores the impact of the 4 different gamma rate categories (gray bar on top of each panel) in our choice of substitution model for IQTree (GTR+F+I+G4). Rate is on the Y-axis, and the different mixture datasets are ordered from *pol* to wgs100 on the X axis. Boxes in panels indicate the central 50% (top and bottom of boxes) and the median (thicker black line in boxes) of the proportions, whiskers indicate the range, and dots indicate outliers. There is a monotonic relationship between relative rate values and wgs proportion beginning with wgs10, when wgs sequences are introduced.
(TIFF)

## Acknowledgments

Part of this research was conducted using computational resources and services at the Center for Computation and Visualization (CCV), Brown University. Thank you to Ashok Ragavendran, Joselynn Wallace, Eric Salomaki, and the rest of the CCV team for feedback and suggestions.

## Author contributions

**Conceptualization:** August Guang, Casey W. Dunn, Vlad Novitsky, Mark Howison, Rami Kantor.

**Data curation:** August Guang.

**Formal analysis:** August Guang.

**Funding acquisition:** Rami Kantor.

**Investigation:** August Guang.

**Methodology:** August Guang, Casey W. Dunn.

**Project administration:** August Guang, Casey W. Dunn, Rami Kantor.

**Resources:** August Guang, Casey W. Dunn.

**Software:** August Guang.

**Supervision:** Casey W. Dunn, Rami Kantor.

**Visualization:** August Guang.

**Writing – original draft:** August Guang.

**Writing – review & editing:** August Guang, Casey W. Dunn, Vlad Novitsky, Mark Howison, Rami Kantor.

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
