## [Decision Letter · Decision Letter 0]

31 Mar 2025

PCOMPBIOL-D-25-00292

T-shaped alignments integrating HIV-1 near full-length genome and partial pol sequences can improve phylodynamic inference of transmission clusters

PLOS Computational Biology

Dear Dr. Guang,

Thank you for submitting your manuscript to PLOS Computational Biology. After careful consideration, we feel that it has merit but does not fully meet PLOS Computational Biology's publication criteria as it currently stands. Therefore, we invite you to submit a revised version of the manuscript that addresses the points raised during the review process.

Please submit your revised manuscript within 60 days. If you will need more time than this to complete your revisions, please reply to this message or contact the journal office at ploscompbiol@plos.org. Please include the following items when submitting your revised manuscript:

We look forward to receiving your revised manuscript.

Kind regards,

Katharina Kusejko

Academic Editor

PLOS Computational Biology

Roger Kouyos

Section Editor

PLOS Computational Biology

**Additional Editor Comments (if provided):**

The two reviewers have raised several important points that should be addressed. In addition, I have the following comments:

1) In the "Author Summary", the authors write that a novel approach for analyzing the spread of HIV-1 is explored. This is somewhat misleading, as this manuscript does not look into HIV-1 transmission, but clustering results in a convenience samples. No conclusions about any HIV transmission dynamics can be drawn here. Similar, the lines 226-229 in the Discussion concerning public health go in my opinion far beyond what was done here, a recommendation concerning cluster certainty should be avoided.

2) For this manuscript, as this describes mainly the workflow for exploring the impact of different fractions of full-length genomes, having the "Methods" section before the "Results" section would improve the reading flow.

3) Methods: To me it is not clear - also Reviewer 1 pointed this out - what was shuffled 1000 times randomly. This is a crucial point: Did you shuffle the order of sequences every time, and thus different sequences were "cropped" to consist of pol only? I am worried that if this was not shuffled properly, and thus always the same sequences were cropped, this might lead to artificial clustering due to sequences in the Los Alamos Database being more similar. Please clarify this aspect in the discussion. You might also think about a "methods" Figure clearly explaining the algorithm and different measures you applied to assess the impact of the fraction of full-genomes.

4) Selection of all available Los Alamos sequences: As these sequences stem from different studies and techniques (as Reviewer 2 pointed out), it would be interesting to see a sensitivity analysis looking into a subset of the sequences stemming from one study only (i.e., you could take the largest sub-study with available wgs). In my opinion, with this quite biased selection of available sequences, it is not clear which phylogenetic patterns would be expected at all - the current selection of sequences do not belong to one sub-epidemic in a country/region/time-frame/sub-population/etc. Thus the results are also hard to interpret.

5) Cluster Definition: You defined clusters as matching only if all members were the same. This is a very strict definition, and in my opinion might be too strict. E.g., if you have a cluster with 10 members, even if this cluster is still there with only one member difference, this would lead to a "non-match". I suggest to at least run a sensitivity analysis applying other cluster definition, e.g. more than 50%, or more than 2/3 of the members being the same. The result presented in Figure 4 might be a consequence of this strict definition: It is concerning that already when comparing all wgs to wgs90, the AMI is only between 0.3 to 0.5. If the AMI is anyway quite low, does the difference between the lower wgs values (wgs90, wgs80, wgs70,...) matter so much? It seems that even 10% missing wgs is leading to a much worse result. To understand this better: could you include results of wgs99, wgs95 or other intermediate values as well to understand this "jump" at the beginning? This also questions the whole "take-home message" that at least 50% wgs is beneficial: the actual changes in AMI are minimal.

6) Discussion: please do not use the term "phylodynamic" here, as no dynamic aspect (e.g. cluster growth) was investigated

**Journal Requirements:**

2) We noticed that you used the phrase 'data not shown' in the manuscript. We do not allow these references, as the PLOS data access policy requires that all data be either published with the manuscript or made available in a publicly accessible database. Please amend the supplementary material to include the referenced data or remove the references.

3) Please ensure that the funders and grant numbers match between the Financial Disclosure field and the Funding Information tab in your submission form. Note that the funders must be provided in the same order in both places as well.

**Reviewers' comments:**

Reviewer's Responses to Questions

**Comments to the Authors:**

Reviewer #1: The study by Guang et al assessed the impact of having sequences with varying length on clustering analysis, or more specific how much improvement including full-length sequences can have on the overall clustering outputs. This is an important aspect of clustering, particularly when it is used for public health use in HIV, where information on clusters may be used to inform policies. Thus, accuracy is important. Overall, it is a neat study looking at a common problem in sequence analysis. The study design seems sound, though I am not familiar with the Robinson-Foulds distance and the adjusted mutual index. Thus, I cannot comment on the results for these.

Though, this is a computational journal I struggled to fully understand the study and believe the manuscript could benefit from more detail and clarifications. Also, how does the IQTree algorithm deal with missing data, for example does it ignore positions with incomplete coverage? I may have missed it but some explanations on that would be helpful. Apologies if I missed this, but did including wgs improve bootstrapping support overall?

Minor comments:

Lines 74 to 79 could be re-written to clarify the outcomes. I didn’t understand fully what the distance along y-axis represents and what the distance. I understand how the points move along the axis from wgs20 to wgs90 but not why pol and wgs10 are ‘off’ the line, as in wgs10 is worse than pol?

Lines 253 to 260 not very clear

Line 144. Replace ‘developed’ with ‘assessed’ or similar as per my understanding the study did not ‘develop’ the approach, rather compared outputs.

Line 244. While sampling dates are important, I don’t think we can stay bootstrap and genetic distances are ‘arbitrary’. I would rephrase this.

Line 245. Unclear what exactly was repeated 1000 times? The clustering? The generation of mixed length datasets?

Figures

Figure 1. It seems the more ‘wgs’ sequences are present the more variation is in the distance comparison (larger confidence intervals), why is that?

Figure 3 is a bit chaos. Can you separate by bootstrap? Make each panel small, I think that should be sufficient.

Reviewer #2: See attachment.

**Have the authors made all data and (if applicable) computational code underlying the findings in their manuscript fully available?**

Reviewer #1: None

Reviewer #2: Yes

PLOS authors have the option to publish the peer review history of their article (what does this mean?). If published, this will include your full peer review and any attached files.

Reviewer #1: No

Reviewer #2: No

**Figure resubmission:**
---

## [Decision Letter · Decision Letter 1]

9 Oct 2025

PCOMPBIOL-D-25-00292R1

T-shaped alignments integrating HIV-1 near full-length genome and partial pol sequences can improve phylogenetic inference of transmission clusters

PLOS Computational Biology

Dear Dr. Guang,

Thank you for submitting your manuscript to PLOS Computational Biology. After careful consideration, we feel that it has merit but does not fully meet PLOS Computational Biology's publication criteria as it currently stands. Therefore, we invite you to submit a revised version of the manuscript that addresses the points raised during the review process.

Please submit your revised manuscript within 30 days Dec 09 2025 11:59PM. If you will need more time than this to complete your revisions, please reply to this message or contact the journal office at ploscompbiol@plos.org. Please include the following items when submitting your revised manuscript:

We look forward to receiving your revised manuscript.

Kind regards,

Katharina Kusejko

Academic Editor

PLOS Computational Biology

Roger Kouyos

Section Editor

PLOS Computational Biology

**Journal Requirements:**

1) Thank you for stating "Publicly available data at https://www.hiv.lanl.gov/content/sequence/HIV/mainpage.html was used for the analysis." Please confirm whether all data needed to replicate the study's findings are at this link? If so, please update your Data Availability statement to "All data required to replicate the study's findings can be found here  https://www.hiv.lanl.gov/content/sequence/HIV/mainpage.html

**Reviewers' comments:**

Reviewer's Responses to Questions

Reviewer #2: This manuscript has improved significantly through all additional analyses and points discussed. Although the study is very thoroughly conducted and has gained improvements, the setup reflects a highly experimental scenario with many uncertainties. Thus, I would suggest to position the attitude towards public health implementations and recommendations for incorporating such nFL genomes even more neutral (e.g. L278-287).

Minor points to address:

- L 106/107: You have two unfinished sentences regarding the overlap coefficient.

**Have the authors made all data and (if applicable) computational code underlying the findings in their manuscript fully available?**

Reviewer #2: Yes

PLOS authors have the option to publish the peer review history of their article (what does this mean?). If published, this will include your full peer review and any attached files.

Reviewer #2: No

**Figure resubmission:**
---

## [Editor Report · Decision Letter 2]

29 Oct 2025

Dear Dr. Guang,

We are pleased to inform you that your manuscript 'T-shaped alignments integrating HIV-1 near full-length genome and partial pol sequences can improve phylogenetic inference of transmission clusters' has been provisionally accepted for publication in PLOS Computational Biology.

Best regards,

Katharina Kusejko

Academic Editor

PLOS Computational Biology

Roger Kouyos

Section Editor

PLOS Computational Biology

---

## [Editor Report · Acceptance letter]

PCOMPBIOL-D-25-00292R2

T-shaped alignments integrating HIV-1 near full-length genome and partial pol sequences can improve phylogenetic inference of transmission clusters

Dear Dr Guang,

I am pleased to inform you that your manuscript has been formally accepted for publication in PLOS Computational Biology. Your manuscript is now with our production department and you will be notified of the publication date in due course.

With kind regards,

Zsofia Freund
